# An Investigation of the Association between 3D Spinal Alignment and Fibromyalgia

**DOI:** 10.3390/jcm12010218

**Published:** 2022-12-28

**Authors:** Amal Ahbouch, Ibrahim M. Moustafa, Tamer Shousha, Ashokan Arumugam, Paul Oakley, Deed E. Harrison

**Affiliations:** 1Department of Physiotherapy, College of Health Sciences, University of Sharjah, Sharjah 27272, United Arab Emirates; 2Neuromusculoskeletal Rehabilitation Research Group, RIMHS–Research Institute of Medical and Health Sciences, University of Sharjah, Sharjah 27272, United Arab Emirates; 3Faculty of Physical Therapy, Cairo University, Giza 12613, Egypt; 4CBP Nonprofit (a Spine Research Foundation), Eagle, ID 83616, USA; 5Private Practice, Newmarket, ON L3Y 8Y8, Canada; 6Kinesiology and Health Sciences, York University, Toronto, ON M3J 1P3, Canada

**Keywords:** fibromyalgia, posture, prediction, regression analysis, formetric analysis

## Abstract

Fibromyalgia syndrome (FMS) is a common condition lacking strong diagnostic criteria; these criteria continue to evolve as more and more studies are performed to explore it. This investigation sought to identify whether participants with FMS have more frequent and larger postural/spinal displacements in comparison to a matched control group without the condition of FMS. A total of 67 adults (55 females) out of 380 participants with FMS were recruited. Participants with FMS were sex- and age-matched with 67 asymptomatic participants (controls) without FMS. We used a three-dimensional (3D) postural assessment device (Formetric system) to analyze five posture variables in each participant in both groups: (1) thoracic kyphotic angle, (2) trunk imbalance, (3) trunk inclination, (4) lumbar lordotic angle, and (5) vertebral rotation. In order to determine whether 3D postural measures could predict the likelihood of a participant having FMS, we applied the matched-pairs binary logistic regression analysis. The 3D posture measures identified statistically and clinically significant differences between the FMS and control groups for each of the five posture variables measured (*p* < 0.001). For three out of five posture measurements assessed, the binary logistic regression identified there was an increased probability of having FMS with an increased: (1) thoracic kyphotic angle proportional odds ratio [Prop OR] = 1.76 (95% CI = 1.03, 3.02); (2) sagittal imbalance Prop OR = 1.54 (95% CI = 0.973, 2.459); and (3) surface rotation Prop OR = 7.9 (95% CI = 1.494, 41.97). We identified no significant probability of having FMS for the following two postural measurements: (1) coronal balance (*p* = 0.50) and (2) lumbar lordotic angle (*p* = 0.10). Our study’s findings suggest there is a strong relationship between 3D spinal misalignment and the diagnosis of FMS. In fact, our results support that thoracic kyphotic angle, sagittal imbalance, and surface rotation are predictors of having FMS.

## 1. Introduction

Fibromyalgia syndrome (FMS) is a prevalent musculoskeletal condition that manifests with pain, stiffness, and tenderness of different body structures, such as muscles and tendons. Characteristic symptoms of FMS include general malaise with anxiety and depression, poor sleep, cognitive dysfunction, and disturbances in bowel functions [1,2]. FMS is underdiagnosed and undertreated due to the complexity of the multiple symptoms and comorbidities associated with it [3]. Despite the many efforts that have been made with specific diagnostic criteria for FMS, healthcare providers still find these criteria unclear, which causes a lack of confidence when using them [3,4]. In fact, these criteria have required multiple revisions as more and more data were made available. Despite elimination of the associated symptoms criterion in the 1990 classification and the tender points examination in the 2010/2011 revision, the 2016 revision included these two criteria [4]. 

Despite the debate around the FMS diagnostic criteria, it is well-established that pain is a centrally mediated phenomenon [2,4,5,6]. Currently, the evidence conceptualizes pain as the personal experience of a complex process compiling sources of input from joint mechano-receptors and information from the general environment, coupled with previous painful experiences and or memories of a painful event [6]. In fact, evidence has linked pain to posture and supported that moderation or elimination of pain can be achieved through improved posture [7,8]. While this association between pain and postural alignment has been questioned by other studies [9,10], the evidence supporting it is constantly growing [7]. For example, evidence supports a strong link between body misalignment and pain syndromes, many of which are found in FMS patients, including: tension-type and cervico-genic headaches [11,12], temporomandibular disorders [13,14,15], shoulder impingement [16,17], abnormal sagittal plane postures like protracted shoulders and forward head posture, respiratory dysfunction [18,19], back pain [20,21], impaired balance [22], FMS itself [23], and osteoporotic spinal deformity [24]. 

Determining objective postural assessment outcome measures could add another dimension to the diagnostic criteria of FMS, leaving less room for doubt for healthcare providers in diagnosis, and guiding more robust interventional treatments. The objective of the current investigation is to examine the potential relationship between FMS and postural misalignment, through detailed measurement of three-dimensional (3D) posture including kyphotic and lordotic angles, sagittal and coronal imbalance, and vertebral rotation. We designed a case control investigation to explore any postural diagnostic relationships that might predict cases with FMS versus those controls without FMS in an effort to help with diagnosing and treating this complex condition. The current study explores the hypothesis that 3D postural analysis will be able to accurately determine FMS participants compared to matched healthy controls.

## 2. Materials and Methods

We used a single-blind case control design to assess possible differences in 3D postural alignment among participants with chronic FMS in comparison to an age- and sex-matched asymptomatic control group. All ethical standards for use in human experimental research designs were followed in compliance with the Helsinki Declaration. Participant recruitment began following an approval from the Cairo University Ethics Committee (approval number: Cairo-PT.3-4561). Participants were recruited via advertisements posted on notice boards and relevant social media pages. 

Initial inclusion criteria were assessed via a screening phone call for potential participants, and those with suspected FMS were then scheduled for and received a detailed evaluation with one of three neurologists working at our outpatient department to confirm their eligibility to participate. The asymptomatic control group comprised of volunteers who received the same examination and assessments using a therapist who was blind to the participants’ possible group status (control vs. FMS). In order to be eligible for the control group, participants were required to be asymptomatic and could not report pain during the physical examination process. All participants signed informed consent forms prior to entering the investigation and also prior to data collection.

Sixty-seven adult participants with FMS (≥18 years of age, 12 males and 55 females) out of 380 participants were enrolled after meeting the 2016 fibromyalgia diagnostic criteria [4]. All participants were screened for conditions that would affect their inclusion into our study: severe cardiopulmonary disease and hyper-tension, long-standing viral infections, a history of any significant medical condition, any moderate or severe scoliosis, and a history of spine surgery. 

### 2.1. 3D Posture Measurement Using a Formetric System 

The Formetric software system (DIERS Medical Systems, Chicago, IL, USA) was used to provide analysis of the following posture profiles in three planes: (1) sagittal plane parameters (kyphotic angle, lordotic angle, and sagittal imbalance); (2) frontal/coronal plane parameters (coronal imbalance); (3) and transverse plane parameters (vertebral rotation). This system is both valid and reliable for postural measurements as used herein [25,26]. We followed previously published standard protocols for patient positioning, measurements, and data acquisition for Formetric software analysis, and we refer the interested reader to this publication [12]. A sample Formetric system report is shown in Figure 1 and each of these measurement variables is described below.

### 2.2. Kyphotic Angle

The thoracic kyphotic angle (cervicothoracic transition point (ICT)- thoracolumbar transition point (ITL) max) was measured between tangents from the cervicothoracic junction (ICT-T1) and that of the thoracolumbar junction (ITL-T12). The cutoff value to determine hyper-kyphosis of our participants was set at an angle greater than or equal to 55° [25,27]. Formetric measurements of thoracic kyphosis over-estimate the actual radiographic measured kyphosis of the same person by a mean of 7–8°; however, a strong correlation between the two different measurement methods has been found (r = 0.79 to 0.872) [25,27]. Thus, a formetric value of 55° for thoracic kyphosis would approximate a 48° radiographic measurement value from T1–T12 (the upper end of normal in healthy middle-aged adults) [28].

### 2.3. Lordotic Angle

The lumbar lordotic angle was assessed between the intersection of two lines drawn tangent to: (1) a surface marker at the inflection point of the thoraco-lumbar junction (termed ITL) and (2) the point of inflection between the lumbar and sacral spines (termed ILS) and the maximum lumbar lordosis was thus termed ITL-ILS max. The Formetric measurement of lumbar lordosis is both reliable and valid with a good correlation (r > 0.70) to radiography and small measurement differences (8° difference from radiographic measurements) [25,27].

### 2.4. Sagittal Imbalance or Trunk Inclination

Sagittal imbalance was measured as a height difference between the vertebral prominence of C7 (VP) and dimple middle (DM), defined as the point lying on the center of the straight line connecting the left dimple to the right dimple, based on a vertical plane (sagittal section). When the VP is anterior relative to the DM, then the angle has a positive value, while if the VP is posterior to the DM, then the angle has a negative value.

### 2.5. Coronal Imbalance

The coronal imbalance of the trunk is measured as the left and right displacement of the VP relative to a DM lying in the center of a straight line that connects the left and right dimples. A positive shift is indicated by the VP offset to the right while a negative shift is directed to the left.

### 2.6. Vertebral Rotation

The vertebral rotation is measured as the root mean square (RMS) of the horizontal components of the surface normal relative to a line of symmetry.

### 2.7. Fibromyalgia Impact Questionnaire (FIQ)

Our primary outcome to determine the relationship between 3D posture displacements and FMS was whether or not posture displacements variables would correlate to the fibromyalgia impact questionnaire FIQ score of the FMS participants. The FIQ is a patient questionnaire designed to quantify the impact of FMS on a participant’s current status, their progress or response to intervention. The FIQ is valid and reliable and has 10 subscales that include: physical, day missed from work, ability to perform job duties, well-being, pain intensity, fatigue or malaise, sleep quality, generalized stiffness, and depression and anxiety. The FIQ is scored from 0–100 with greater scores indicating more disability or impairment due to FMS [29].

### 2.8. Sample Size Determination

A pilot study consisting of 10 participants with FMS compared to 10 age- and sex-matched controls was performed, and this data was used to identify the sample size of participants needed for statistical findings. The mean differences and SD of the posture parameters; kyphotic angle, lordotic angle, sagittal imbalance, coronal imbalance; vertebral rotation were: kyphotic angle: −12 (SD 6.2), lordotic angle: –4 (SD 4.8), –sagittal imbalance: –5.2 (SD 2.1) coronal imbalance: –4 (SD 1.9), vertebral rotation: –3.9 (SD 1.8). We applied a Bonferroni correction to adjust the significance value for each of the primary outcomes. Using the largest value of the needed sample size estimate determined the final sample size for our trial. It was determined that 56 participants in each group (with a statistical power of 90%) was necessary herein; we increased the sample size by 20% to account for possible participant dropouts.

### 2.9. Data Analysis

In order to test normality of the distribution of our data, we used a one-sample Kolmogorov–Smirnov normality test. Where the data are normally distributed, they are presented as mean ± standard deviation (SD). Baseline participant demographics of age, weight, body mass index (BMI), highest education level completed, marital status, and pain duration, were compared between both groups using the independent t tests for continuous data and chi square tests of independence for categorical data. The Student’s t-test was used to compare the means of continuous variables between the two groups. A *p*-value of < 0.05 was considered statistically significant. A matched-pairs binary logistic regression procedure for estimating odds ratios for a matched pairs case-control design determined whether the 3D posture parameters (kyphotic angle (ICT-ITL (max)), lordotic angle (ITL-ILS (max)), sagittal imbalance (VP-DM), coronal imbalance (VP-DM), and vertebral rotation (RMS)) demonstrated an association with the likelihood of experiencing FMS. Multiple regression was carried out to examine whether posture parameters could significantly predict FMS participants’ FIQ scores. SPSS version 20.0 software was used for analyzing data (IBM SPSS Inc., Armonk, NY, USA).

## 3. Results

We screened by phone greater than 380 possible participants. The most often reason for participant exclusion was an uncontrolled medical condition such as diabetes mellitus, heart disease, renal failure, and so forth. A hierarchy of control group participant was applied whereby a control participant was only included after a FMS group participant of a similar age and gender had been recruited, thus, further exclusions occurred when matching was not possible. Included group participants were: (1) FMS (mean age 46.4 years, SD = 9; 12 males, 55 females) and (2) 67 sex and age matched controls (mean age = 46.5 years, SD = 9.1; 12 males, 55 female). Figure 2 demonstrates the participants’ inclusion and exclusion flow chart for this study.

### 3.1. Sample Characteristics

The baseline participant demographics are presented in Table 1. The FMS and control groups were statistically matched for each of the included demographic variables (Table 1).

### 3.2. Formetric Postural Variables between Group Differences

Each of the five postural variables were found to be statistically significant different between both groups: kyphotic ICT-ITL (max) (*p* < 0.001); lordotic angle ITL-ILS (max) (*p* < 0.001); sagittal imbalance (*p* < 0.001); coronal imbalance (*p* < 0.001); and vertebral rotation (rms) (*p* < 0.001). Table 2 reports the 3D Formetric data means and SD between the FMS and control groups.

### 3.3. Binary Logistic Regressions

The binary logistic regression analysis identified a statistically significant increased probability of having FMS as the following postural variables become increasingly abnormal: (1) kyphotic angle ICT-ITL (max) (*p* < 0.03), (2) sagittal trunk imbalance (*p* = 0.005), and (3) vertebral rotation (rms) (*p* = 0.015). In contrast, no statistically significant differences were found for the two remaining postural displacement variables of coronal balance (*p* = 0.50) and lumbar lordotic angle (ITL-ILS (max); *p* = 0.10). See Table 3.

### 3.4. Odds Ratios between Having FMS and Posture Variables

Three of the five postural displacement variables were found to have statistically significant odds ratios identifying an increased risk of having FMS with increases in the magnitude of the abnormal posture. These three postural variables and their odds ratios were: (1) Thoracic kyphotic angle = 1.76 (95% CI = 1.03, 3.02) indicating that for each degree of increased angle, there was an approximate 76% increased likelihood of having FMS; (2) Sagittal imbalance = 1.54 (95% CI = 0.973, 2.459) indicating that for each degree increase of sagittal imbalance, there was an approximate 54% increased likelihood of having FMS; (3) Surface rotation = 7.9 (95% CI = 1.494, 41.97) indicating that with each 7 degrees increase in surface rotation, there was an approximate 90% increased likelihood of having FMS in this sample.

Similarly, multiple linear regression analysis identified that the 3 postural displacement variables (thoracic kyphotic angle, sagittal imbalance, and vertebral surface rotation) were statistically significant predictors of a participant’s FMS impact questionnaire scores; F = 104.4, *p* < 0.01. The multiple-regression analysis revealed that 80% of the variance in the dependent variable FMS impact questionnaire scores could be explained by the independent variables (postural displacements). Table 4 presents these findings.

## 4. Discussion

This case control investigation sought to identify if 3D posture displacements can identify FMS participants versus a matched control group without overt signs and symptoms and no FMS. The results identified from this study show that there is a strong relationship between spinal 3D misalignment and the diagnosis of FMS. In fact, our results support that the outcome measures used to objectively assess thoraco-lumbar postural alignments could be used as strong predictors for the diagnosis of FMS. We believe that our investigation is the first study to seek and identify a predictive association between comprehensive 3D thoraco-lumbar postural alignments and FMS.

Some recent studies investigated spine posture in individuals, commonly women, with FMS [30,31,32,33]. Sempere-Rubio et al. [30] found that there is an altered trunk posture in women with FMS compared to a control group. These findings agree with those of the current study; both studies found differences in the same direction for one of the most common outcome measures; namely an increased kyphotic angle of the group with FMS [30]. However, our study offers other more objective outcome measures for the posture assessment by using the 3D Diers Formetric device. Another study by Sempere-Rubio et al. showed that women with FMS have an altered postural control compared to a healthy group, adding to the strong factors that should be considered in diagnosing and treating FMS [31].

The increased thoracic kyphosis in FMS populations has also been supported by Celenay et al. [33], who found that women with FMS had an increased thoracic kyphosis angle compared to the control group. However, the lumbar lordosis was not significantly different between groups while in our study the group with FMS had a slightly higher lumbar lordotic angle compared to the control group. This discrepancy for lumbar lordosis between investigations is likely explained by the fact that the compensation to an increased kyphotic angle (which is the confirmed common deviation in all the studies mentioned including our study) can be either an increased or a decreased lordotic angle depending on the general posture assumed by the individual [34]. The increased kyphosis in the kyphotic, flat-back and sway-back postures is linked to neutral lumbar spine, hypolordotic upper lumbar spine and a hyperlordotic lower lumbar spine in each of those three postures, respectively [34].

Although a normal clinical neurological examination is most often identified, FMS patients consistently report sensory deficits which has been confirmed on dynamic posturography [35,36]. Indeed, abnormalities of body posture in women suffering from FMS have been identified to be related to poor trunk position sense and balance instability [33]. Sempere-Rubio et al. identified that females with FMS have poorer postural control compared to a healthy group which further emphasizes including postural alignment in diagnosing and treating FMS [31]. Another study by Sempere-Rubio et al. reported that a decreased ability in maintaining sitting thoracic posture could predict a reduction in quality of life in women with FMS [32].

Previously, investigators have suggested that objective studies are needed to understand postural balance abnormalities in FMS populations and their relationship to different types of musculoskeletal abnormalities [37]. The relationship between FMS and postural misalignment has been investigated in several studies [7,38,39,40]. Moustafa and Diab speculated that sustained postural imbalances can result in the establishment of a state of continuous asymmetric loading [38]. If a significant maladaptive posture is sustained long enough, it will consequently affect the quality of life [41]. These speculations are supported by other authors [42,43] who discuss that biomechanical dysfunction causing a sustained asymmetrical loading and muscle imbalance leads to an increased stress and strain on body structures. For instance, Hiemeyer et al. linked poor flexed posture to the tender points characterizing FMS, most of which are in postural muscles [43]; restoring a correct posture diminished these tender points if not completely eliminated them [43,44].

The most intriguing finding was that the predictors of postural features included not only the sagittal profile, but also the abnormal transverse profile as indicated by surface rotation. This finding is not surprising as previously, Veldhuizen et al. [45] identified that alterations in transverse plane rotational alignment were positively correlated to sagittal alignment abnormalities. Similarly, other authors have reported the correlation between the sagittal and coronal and transverse spinal contours [46,47,48]. Incorrect posture and spinal abnormalities in sagittal and axial planes may modify the stability of this structure and its load distribution, which can generate abnormal stresses and strains, thus provoking a reduction in quality of life and an increased risk of FMS.

The present paper adds to the present body of FMS literature that supports an optimal alignment of upright human posture. Due to the significant relationships between the magnitude of the posture displacements as identified with the Formetric analysis and the odds of having FMS our findings indicate that postural displacements are predictive of not only who has fibromyalgia but also of the severity of the identified disability as measured with the FIQ. Our findings are in general agreement of the randomized trial by Moustafa where it was identified that correction of the cervical sagittal plane alignment was found to improve 3D posture and improve the short- and long-term pain and impairments of patients suffering from chronic FMS [44].

### 4.1. Limitations

Because this study was a cross-sectional, case-control investigation and not a treatment outcome trial, it is unknown how our findings may or may not influence patient treatment outcomes when postural rehabilitation is pursued. However, since the 3D postural analysis showed increased posture aberrations in the FMS group, we recommend that interventions designed to improve 3D posture should be implemented as part of a multi-modal treatment approach. Lastly, although the Formetric measurement of 3D posture is valid and reliable [25,27], it does not completely describe the actual sagittal and coronal spine alignment. In FMS participants, using sagittal and anterior-posterior radiological profiles would likely give further insights into exact rotation and translation displacements of individual vertebral and overall spine curvature geometry and magnitudes.

### 4.2. Conclusions

The results derived from this study identified that there is a strong relationship between spinal three-dimensional (3D) misalignment as measured with the Formetric system and the diagnosis of FMS. Our results support that the posture displacements of the thoraco-lumbar regions can be used as part of the clinical and diagnostic indicators to determine who has FMS and as possible outcomes of treatment interventions. Future trials should use 3D postures as an outcome measure to determine if posture rehabilitation impacts short- and long-term outcomes in FMS sufferers.

## Figures and Tables

**Figure 1 jcm-12-00218-f001:**
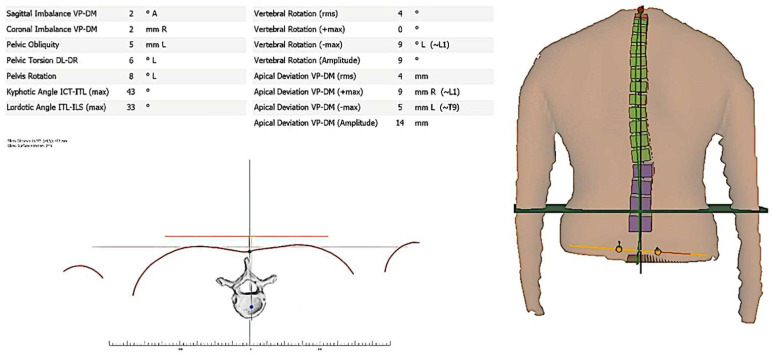
Illustrative example for the Formetric report.

**Figure 2 jcm-12-00218-f002:**
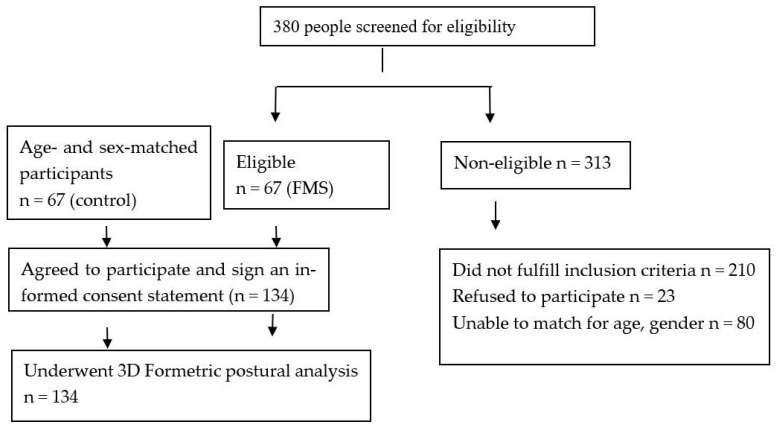
Participant flow chart.

**Table 1 jcm-12-00218-t001:** Baseline participant demographics.

Variable	Fibromyalgia Group (*n* = 67)	Control Group (*n* = 67)
Age (years)	46.4 ± 9	46.5 ± 9.1
Weight (kg)	75 ± 9	80 ± 10
Gender (%)		
Male	12	12
Female	55	55
Body mass index mean (SD), Kg/m^2^		
Graduation		
Primary school	5 (7.5%)	2 (3%)
Secondary school	10 (14.9%)	8 (11.9%)
Advanced technical colleague certificate	10 (14.9%)	15 (22.4%)
University diploma	32 (47.8%)	30 (44.8%)
Others	10 (14.9%)	12 (17.9%)
Marital status (%)		
Single	5 (7.5%)	4 (6%)
Married	55 (82.1%)	57 (85%)
Separated, divorced, or widowed	7 (10.4%)	6 (9%)
Pain duration		
1–5 y	20 (29.9%)	Asymptomatic
>5 y	47 (70.1%)	Asymptomatic

There were no statistically significant differences between the FMS and control groups; *p* > 0.05 for all variables using the independent t test for continuous data and chi squared test of independence for categorical data. y: year.

**Table 2 jcm-12-00218-t002:** Means, standard deviation (SD), 95% confidence interval (CI) and statistical significance of the postural measurements between participants with FMS and controls.

3D Formetric Measurement		Mean	±SD	SEM	Cohen’s *d*	95% CI	*p*-Value
Kyphotic angle ICT-ITL (max) (deg.)	FMS	74.1	4.75	0.58	6.8	[14.3–19.04]	<0.001
	Control	57.4	8.41	1.02			
Lordotic angle ITL-ILS (max) (deg.)	FMS	45.1	5.71	0.69	4.3	[1.9–4.9]	<0.001
	Control	41.5	2.46	0.30			
Sagittal imbalance (mm)	FMS	9.53	2.77	0.33	2.3	[4.1–5.7]	<0.001
	Control	4.5	1.89	0.23			
Coronal imbalance (mm)	FMS	8.04	3.19	0.39	2.4	[3.6–5.6]	<0.001
	Control	3.22	1.37	0.16			
vertebral rotation (rms) (deg.)	FMS	9.5	1.86	0.22	1.8	[3.5–4.7]	<0.001
	Control	5.3	1.75	0.21			

SEM: Std. Error of Mean; SD: standard deviation; CI: confidence interval; ICT-ITL: Cervico-thoracic inflection point-thoraco-lumbar inflection point; ITL-ILS: thoracic-lumbar inflection point- lumbo-sacral inflection point; rms: root mean square.

**Table 3 jcm-12-00218-t003:** Variables in the equation for logistic regression and odds ratio calculation.

Variables in the Equation								
	B	S.E.	Wald	df	Sig.	Exp(B)	95% CI for EXP(B)	
							Lower	Upper
Sagittal imbalance (mm)	0.437	0.236	3.409	1	0.005	1.547	0.973	2.459
vertebral rotation (rms) (degrees)	2.069	0.851	5.914	1	0.015	7.919	1.494	41.970
Kyphotic ICT−ITL (max) (degrees)	0.569	0.275	4.273	1	0.039	1.766	1.030	3.029
Coronal imbalance (mm)	−0.188	0.313	0.360	1	0.549	0.829	0.449	1.530
Lordotic angle (degrees)	0.472	0.326	2.105	1	0.147	1.604	0.847	3.035
Constant	−52.309	21.782	5.767	1	0.016	0.000		

Variable(s) entered on step 1: Sagittal imbalance, vertebral rotation, Kyphotic ICT-ITL (max), Coronal imbalance, Lordotic angle.

**Table 4 jcm-12-00218-t004:** Multiple linear regression for the fibromyalgia impact questionnaire score versus postural variables and their associated risk factors.

Model		Unstandardized Coefficients		Standardized Coefficients	t	Sig.	95.0% Confidence Interval for B	
		B	Std. Error	Beta			Lower Bound	Upper Bound
1	Constant	−2.043	0.214		−9.562	<0.001	−2.466	−1.620
	Kyphotic ICT−ITL (max) (deg.)	0.021	0.003	0.459	8.462	<0.001	0.016	0.026
	sagittal imbalance (mm)	0.024	0.009	0.166	2.739	0.005	0.007	0.042
	vertebral rotation (deg.)	0.078	0.011	0.430	7.197	<0.001	0.057	0.099
	Lordotic angle (deg.)	0.010	0.004	0.092	2.206	0.029	0.001	0.019
	Coronal imbalance (mm)	−0.007	0.010	−0.046	−0.688	0.493	−0.026	0.013

B: Unstandardized coefficients.

## Data Availability

The datasets analysed in the current study are available from the corresponding author on reasonable request.

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
