# Peer review of "An Investigation of the Association between 3D Spinal Alignment and Fibromyalgia"

_jcm, 2022, doi:10.3390/jcm12010218_

Round 1
Reviewer 1 Report
To authors,
Thank you for your hard work for this publication. It is interesting to find out new insights for FMS regarding spinal alignment and posture.
Through this study, authors have tried to figure out the new objective diagnostic criteria for FMS with the novel method of measuring their postures. This study suggested that the sagittal imbalance including kyphotic angle and apical rotation have increased likelihood of having FMS although the coronal balance did not have an impact for the likelihood of FMS.
It is an interesting article but there are still come concerns for this method and conclusions. Overall, it is too conclusive to mention about diagnosis by objective posture measurements. Below you can find my comments:
Abstract
Line 20-21: In which part of spine does kyphotic angle means? thoracic, lumbar or whole spine? Regarding, trunk imbalance, trunk inclination, lordotic angle and vertebral rotation should be detailed because of the novelty of 3D Formetric device assessing the posture.
Line24-26: It would be nice to address odds ratio of each parameters also in abstract, which is important information for readers to know how much important which parameters impact on the diagnosis.
Line 29-31: The conclusion of strong predictors is too conclusive. These objective measurement of posture can be a hint for diagnosing the chronic pain. However, you need to discuss about the cutoffs of ROC curve and AUC.
Materials and Methods
Line 71: Who were blinded for this study?
Line 78-80: As you mentioned in introduction section, the diagnosis of FMS is complicated. The diagnosis by neurologist is great but how did they rule out other disorders associated with chronic pain and posture change such as adult spinal deformity, major depression and etc.?
Line 83: Are asymptomatic participants volunteers?
Line 89: Why do you exclude these internal medicine problems such as hepatitis, cardiopulmonary problems, unstable hypertension, polio, and epilepsy?
Line 92-93: It is better to move the sentence "Figure 1 demonstrates..." and figure of flow chart into results section because of redundancy.
Line 122: The patient included the patient only or all the participants?
Line 126-127: Are these angles for thoracic, lumbar or whole spine? Please be specific.
Line 141-143: It's really minor but does ICT mean "inter-cervico-thoracic point"? ITL as well?
Line 153: Dose >0.700 indicate correlation coefficient? Please align the description with Line 145. If so, P-value is not needed.
Line 157-160: You mentioned this trunk inclination is an angle in line 158-160. Sagittal imbalance is estimated from the trunk inclination, indicated by the distance between the sacral prominence and plumb line from C7 spinous process prominence. Is that correct? Could you make them more specific to read? and what does the dimple middle mention about?
Line 162: VP meaning vertebral prominence and DM was mentioned already in the last section. Which level of vertebrae does the VP here mean?
Line 169: Does FIQ have any cutoff line for diagnosing FMS? or just assess the QOL in patients with FMS?
Line 181-184: Was this mean differences and SD of posture parameters were measured from healthy participants or patients with FMS?
Line 185-190: This study seems to be a retrospective comparative study because you screened potential 380 participants first. Thus the sample determination should be done with post-hoc manner to calculate the power.
Line 196: what do you mean by "matched-pairs" binary logistic regression?
Results
Line 215 (Table1):
Please address how to analyze categorical data and it would be great the each P-value of the comparisons.
I'd like to know why you pick up "Graduation" and "Marital status" as demographics. Does it affect the status of FMS?
Line 229-230, 234 (Table3): The sagittal imbalance with P = 0.05 and odds ratio CI 0.9-2.459 should be interpreted as non-significant because you mentioned the P < 0.05 is significant. Please align the expression with others by writing the third decimal place, and please explain for this interpretation.
Discussion
Line 271: Does "curvature" indicate kyphosis?
Line283-290: The relationship between postural control and FMS is interesting but your great work revealed the vertebral rotation can impact the most for the QOL in patients with FMS. Is there any discussion for this result?
Line 292-300: These papers mentioned that the degenerative change of spine can cause pain and this is a different topic from FMS. In my understanding, these specific spinal problems should be ruled-out because the concept of FMS is non-specific chronic pain.
Line 307-310: The result of multiple regression of FIQ and posture indicated that just that patients with FMS who have poorer posture have a poorer quality of life? These patients can associated with spinal diseases (especially scoliosis with vertebral rotation) .
Reviewer 2 Report
Exclusion criteria was amongst others a prior history of back and neck surgery. However, this doesn't mean that your cohort didn't have an underlying spinal pathology which is causing the alterations in your 3D scan.
Exclusion criteria should therefor include spinal pathology which should have been worked up prior (clinical exam, MRI scan).
Round 2
Reviewer 1 Report
I appreciate authors' great effort for the making a new evidence for the relationship between FMS and spinal disorder.
The revision has been done in an appropriate way.
Reviewer 2 Report
I'm still missing the exclusion of an underlying spinal pathology.